

# Development of a COVID-19 risk assessment model for participants at outdoor music festivals: evaluation of the validity and control measure effectiveness based on two actual events in Japan and Spain

Michio Murakami[1,2], Tsukasa Fujita[2], Pinqi Li[2], Seiya Imoto[3] and Tetsuo Yasutaka[2]

[1] Center for Infectious Disease Education and Research (CiDER), Osaka University, Suita, Osaka, Japan
[2] Institute for Geo-Resources and Environment, National Institute of Advanced Industrial Science and Technology (AIST), Tsukuba, Ibaraki, Japan
[3] Division of Health Medical Intelligence, Human Genome Center, The Institute of Medical Science, The University of Tokyo, Minato-ku, Tokyo, Japan

Corresponding author
Michio Murakami,
michio@cider.osaka-u.ac.jp

## ABSTRACT

We developed an environmental exposure model to estimate the coronavirus disease 2019 (COVID-19) risk among participants at outdoor music festivals and validated the model using two real events—one in Japan (Event 1) and one in Spain (Event 2). Furthermore, we considered a hypothetical situation in which Event 1 was held but enhanced measures were implemented to evaluate the extent to which the risk could be reduced by additional infection control measures, such as negative antigen tests on the day of the event, wearing of masks, disinfection of environmental surfaces, and vaccination. Among 7,392 participants, the total number of already- and newly-infected individuals who participated in Event 1 according to the new model was 47.0 (95% uncertainty interval: 12.5–185.5), which is in good agreement with the reported value (45). The risk of infection at Event 2 ($1.98 \times 10^{-2}$; 95% uncertainty interval: $0.55 \times 10^{-2}$–$6.39 \times 10^{-2}$), calculated by the model in this study, was also similar to the estimated value in the previous epidemiological study ($1.25 \times 10^{-2}$). These results for the two events in different countries highlighted the validity of the model. Among the additional control measures in the hypothetical Event 1, vaccination, mask-wearing, and disinfection of surfaces were determined to be effective. Based on the combination of all measures, a 94% risk reduction could be achieved. In addition to setting a benchmark for an acceptable number of newly-infected individuals at the time of an event, the application of this model will enable us to determine whether it is necessary to implement additional measures, limit the number of participants, or refrain from holding an event.

## INTRODUCTION

During the global coronavirus disease 2019 (COVID-19) outbreak, the assessment and management of the infection risk during mass gatherings have become urgent issues (*McCloskey et al., 2020*). One risk assessment method is the epidemiological approach. To date, the COVID-19 infection risk related to events has been assessed using randomized controlled trials (*Revollo et al., 2021*) or observational studies including both events with and without the use of infection control measures such as mask-wearing (*The United Kingdom Government, 2021*). However, in the absence of infection control measures, participating in an event may result in a large number of infected individuals (*i.e.*, clusters) (*Smith et al., 2022*). From an ethical perspective, having studies that actively use events without adequate control measures may not be ideal (*de Vrieze, 2021*). Some recent observational studies analyzed events with adequate control measures to assess the infection rate due to the participation in the events or factors associated with infection risk such as vaccination status (*Sami et al., 2022*; *Suñer et al., 2022*). However, these epidemiologic studies are limited in their ability to assess the extent to which individual or combined infection control measures reduce risk.

To overcome the limitations of existing studies, environmental exposure models may be applied and their effectiveness should be assessed. We previously developed an environmental exposure model to assess the COVID-19 risk among spectators at the opening ceremony of the Tokyo 2020 Olympic Games and evaluated the effectiveness of the implementation of control measures, including mask-wearing, physical distance, ventilation, disinfection, and handwashing (*Murakami et al., 2021*). Additionally, we conducted parametric studies to evaluate the effects of the number of spectators, capacity proportions, and infection prevalence by extending the model to other sporting events (*Yasutaka et al., 2022*). In another study, we evaluated the effects of vaccine-testing packages (*Murakami et al., 2022a*). We confirmed the validity of the model based on the fact that no newly-infected individuals were observed among the participants of professional baseball and soccer events in the fiscal year 2020 (*Yasutaka et al., 2022*); however, this validation has limitations due to the unavailability of active testing after these events. Furthermore, the model has been applied only to events held in Japan. It is expected to examine the validity of the model based on events in different countries and to evaluate the effectiveness of the control measures.

Therefore, in this study, we focused on a cluster outbreak case that occurred during an outdoor music festival event (Event 1) with inadequate infection control measures that was held in Japan. In addition, we also targeted another outdoor music event (Event 2), where control measures including mask-wearing were in place and infection rates due to the event have been reported, held in Spain (*Suñer et al., 2022*). The objectives of this study were as follows: First, we extended the environmental exposure model to assess the COVID-19 risk at music festivals and validate the model by comparing the model estimates at Events 1 and 2 with the actual number of reported infected individuals or estimated infection rate. Second, we evaluated the reduction in infected individuals by applying the developed model to a hypothetical situation in which Event 1 was held with additional or enhanced

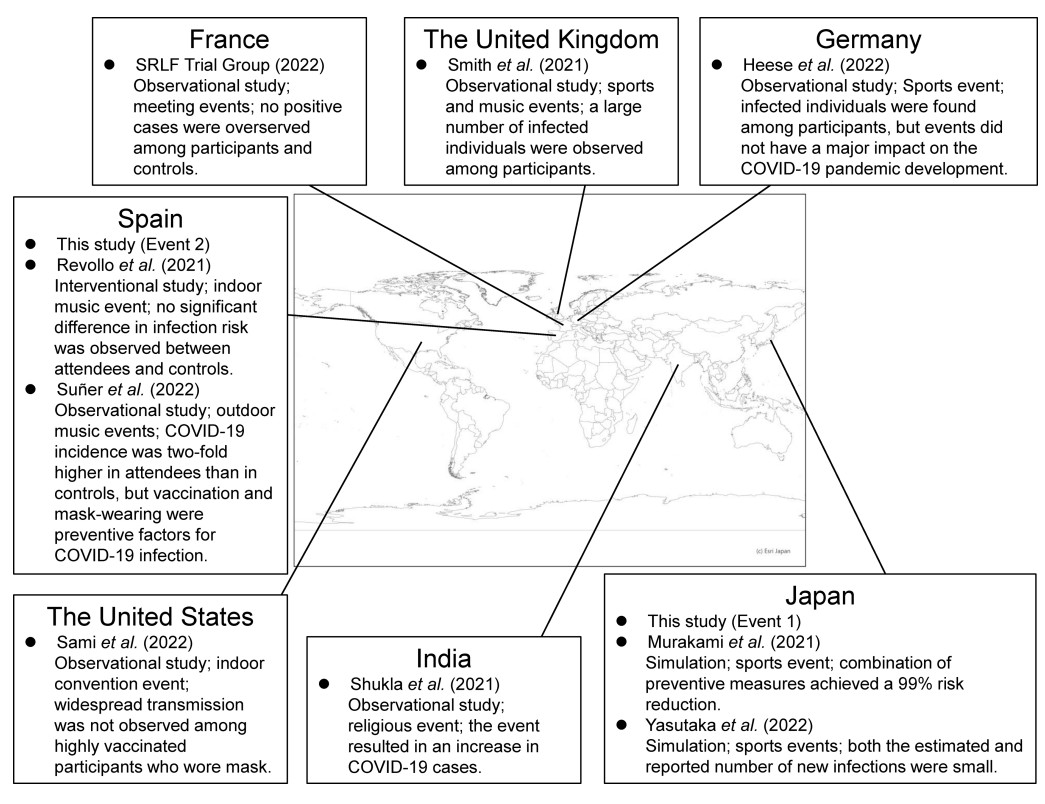

**France**
- SRLF Trial Group (2022) Observational study; meeting events; no positive cases were overserved among participants and controls.

**The United Kingdom**
- Smith *et al.* (2021) Observational study; sports and music events; a large number of infected individuals were observed among participants.

**Germany**
- Heese *et al.* (2022) Observational study; Sports event; infected individuals were found among participants, but events did not have a major impact on the COVID-19 pandemic development.

**Spain**
- This study (Event 2)
- Revollo *et al.* (2021) Interventional study; indoor music event; no significant difference in infection risk was observed between attendees and controls.
- Suñer *et al.* (2022) Observational study; outdoor music events; COVID-19 incidence was two-fold higher in attendees than in controls, but vaccination and mask-wearing were preventive factors for COVID-19 infection.

**The United States**
- Sami *et al.* (2022) Observational study; indoor convention event; widespread transmission was not observed among highly vaccinated participants who wore mask.

**India**
- Shukla *et al.* (2021) Observational study; religious event; the event resulted in an increase in COVID-19 cases.

**Japan**
- This study (Event 1)
- Murakami *et al.* (2021) Simulation; sports event; combination of preventive measures achieved a 99% risk reduction.
- Yasutaka *et al.* (2022) Simulation; sports events; both the estimated and reported number of new infections were small.

**Figure 1** **The locations of target events and other events in previous studies.** The design, type of events, and key findings in previous studies are described (*Heese et al., 2022*; *Murakami et al., 2021*; *Revollo et al., 2021*; *Sami et al., 2022*; *Shukla et al., 2021*; *Smith et al., 2022*; *SRLF Trial Group, 2022*; *Suñer et al., 2022*; *Yasutaka et al., 2022*). The map was created by using Arc GIS (Esri Japan; https://www.esrij.com/).

measures in place. Here, we hypothetically evaluated only Event 1, because our objective was to evaluate the extent to which thorough additional measures would reduce the number of infected individuals. This enabled us to discuss how the application of this model could provide event organizers with a perspective on what additional measures are necessary to limit the emergence of clusters. This is the first study in which an environmental exposure model for the estimation of infection risk was validated using cases with reported infection rates among participants at mass gathering events.

## METHODS

### Event and participants

Two target events were considered in this study. Both events were held during the emergence of the Delta variant. Figure 1 shows the locations of target events and other events in previous studies regarding COVID-19 infection risk at mass gathering events (*Heese et al., 2022*; *Murakami et al., 2021*; *Revollo et al., 2021*; *Sami et al., 2022*; *Shukla et al., 2021*; *Smith et al., 2022*; *SRLF Trial Group, 2022*; *Suñer et al., 2022*; *Yasutaka et al., 2022*).

### Event 1

The first event (Event 1) was Namimonogatari2021, an outdoor hip hop festival held at the Aichi Sky Expo (35,000 m$^2$) in the Aichi Prefecture in Japan from 9:00 to 21:00 (JST) on August 29, 2021 (*Aichi Prefecture, 2021c*). In total, 7,392 people attended the festival and 45 infected individuals were reported (*Aichi Prefecture, 2021c*). Of the participants, 1,154 were tested using the free polymerase chain reaction (PCR) tests that were conducted in the Aichi Prefecture and Nagoya City. As of September 13, 2021, the result of a total of 658 tests were known and included eight positive cases. The total reported number of infected cases (*i.e.*, 45) included infected individuals identified in other areas (*Asahi , 2021*).

The reported number of infected people in the Aichi Prefecture during the week before this event (August 22–28) was 12,072 (*AichiPrefecture, 2021a*). By dividing by the total population of the Aichi Prefecture (*Aichi Prefecture, 2021b*), the reported number of infected persons per 10 million people was determined to be 2,290 persons per day. Following the methodology of a previous study (*Murakami et al., 2021*), the crude probability of a participant being an infector ($P_0$) is $1.3 \times 10^{-3}$ based on weighting the infectivity time (*He et al., 2020b*) and the proportion of asymptomatic and symptomatic individuals (*He, Yi & Zhu, 2020a*).

### Event 2

The second event (Event 2) was an outdoor music event held in Catalonia, Spain on July 8–10, 2022 (*Suñer et al., 2022*). All the individuals underwent rapid antigen testing and only individuals who tested negative were allowed to participate in the event. The infection rate due to the event and the details of compliance with control measures taken during the event, including the mask-wearing, were reported in a previous study (*Suñer et al., 2022*). The average event time was 12 h per day. In total, 34,518 participants attended the event in a 100,351 m$^2$ area. The infection rate per single day due to participation in the event was $1.25 \times 10^{-2}$. This value was calculated from the infection rate of event participants and control groups, the proportion of people regarding the number of days of event participation, and the odds ratio of infection rate by days of event participation. The reported number of infected persons per 10 million people in the host area was 65,800 per week. The $P_0$ was set at $4.0 \times 10^{-3}$ by taking into account the calculation method used in Event 1 and the exclusion rate of positive individuals by rapid antigen testing (see "Model development") (*Murakami et al., 2022a*).

### Model development
#### *Model briefs: common to both Events 1 and 2*

In this study, we extended a previously established model (*Murakami et al., 2022a*; *Murakami et al., 2021*; *Yasutaka et al., 2022*) to music festivals. Briefly, by considering the actual size of the venue, number of spectators, and $P_0$, we calculated the exposure dose related to the behavioral patterns in the event (see details below) and then estimated the number of infected individuals and infection risk. The number of infected individuals was used as the outcome for Event 1 and the infection risk for Event 2, according to the reports (*Aichi Prefecture, 2021c*; *Suñer et al., 2022*). We assessed the effectiveness of the control
measures on infection risk reduction among the participants by separately calculating the infection risk for scenarios in which the control measures were implemented and those in which they were not. The model was run 10,000 times for each scenario. We used a variety of model parameters according to previous studies (*Murakami et al., 2022a*; *Murakami et al., 2021*; *Yasutaka et al., 2022*).

Regarding the exposure dose, we calculated the viruses emitted by infectors, their environmental behavior, inactivation, and surface transfer. In this model, the virus emission by asymptomatic infectors through talking, coughing, and sneezing is divided into four pathways: direct droplet spray, direct inhalation of inspirable particles, hand contact, and inhalation of respirable particles via air. The viral concentration was calculated after considering the inactivation in the environment and the exposure dose was estimated from several environmental and human behavioral parameters, including the breath volume and the frequency of hand contact with surfaces. Regarding the infection risk calculated from the exposure dose, we used the dose–response equation based on the severe acute respiratory syndrome coronavirus (SARS-CoV) in mice (*Watanabe et al., 2010*) and the proportion of asymptomatic infected individuals in humans (*He, Yi & Zhu, 2020a*), as the equation was established on the basis of a wide range of doses.

Total duration was 12 h per day for both Events 1 and 2. The activities of the music festival participants were categorized into five behavioral patterns, that is, (A) attending live performances (60 min × 6 times); (B) entering, exiting, and resting (50 min × 6 times); (C) using restrooms (2 min ×3 times); (D) ordering at concession stands (1 min × 4 times); and (E) eating (25 min × 2 times); representing a total of 720 min. For each behavioral pattern, the amount of exposure was calculated according to the type of person exposed: (1) people accompanying the infector, (2) people in front of the infector at live performance venues, (3) people exposed in restrooms, (4) people exposed at concession stands, and (5) others. The types and numbers of people exposed are shown in Table 1 and the exposure pathways and doses for each behavioral pattern are shown in Tables 2 and 3.

### Event 1 (base scenario)

Considering the possibility that the Delta-variant strain has a 1,000-fold higher viral load than the wild-type strain (*Li et al., 2022*), we carried out a sensitivity analysis for Event 1 and analyzed the results under conditions in which the concentration of the virus in saliva varied 10-, 100-, and 1,000-fold relative to the wild-type strain. Hereafter, unless otherwise noted, risk assessment was conducted under conditions in which the Delta-variant concentration in saliva was 100-fold relative to the wild-type strain.

In the base scenario (without additional measures) at Event 1, mask-wearing and vaccination were considered. The amount of virus emitted by the infector differs depending on whether the infector wears a mask or not (*Murakami et al., 2022a*). Furthermore, exposed individuals wearing masks have a reduced frequency of contact with facial mucosal membranes (*Murakami et al., 2021*). The mask-wearing proportions of the participants were set as follows: While the mask-wearing proportion among the Japanese public is extremely high (>85%) (*YouGov PLC, 2022*), the target event has been criticized for

**Table 1  Type and number of people exposed.** $P_0$: crude probability of a participant being an infector.

| Type of people exposed | Number of people |
| --- | --- |
| (0) Infectors | This value (X) was estimated from the binomial distribution based on the number of participants (Event 1: 7,392 (base scenario); Event 2: 34,518) and $P_0$ (Event 1: $1.3 \times 10^{-3}$ (base scenario); Event 2: $4.0 \times 10^{-3}$). |
| (1) People accompanying the infector | X ×2 (*Murakami et al., 2021*) |
| (2) People in front of the infector at live performance venues | X ×18 (base scenario: one infector produces three people during one attendance of a live performance; six live performances) X ×6 (distance measure scenario: one infector exposes one person during one attendance of a live performance; six live performances) |
| (3) People exposed in restrooms | X ×45 (one infector exposes 15 people per one restroom use (*Murakami et al., 2021*); three restroom visits) |
| (4) People exposed at concession stands | X ×120 (one infector produces 30 exposed people per one order at a concession stand (*Murakami et al., 2021*); four orders at concession stands) |
| (5) Others | Total number of participants minus the sum of (0)–(4) |

not ensuring that masks were worn (*Aichi Prefecture, 2021c*). Therefore, we conducted a sensitivity analysis in which we assumed that 50% of the participants wore masks (base scenario) and then varied the mask-wearing proportion from 0% to 100% in 10% increments. The participants were divided into mask-wearers and non-wearers according to the mask-wearing proportion and the exposure dose was calculated for each category.

The percentage of people who received two doses of the vaccine was set at 45% based on the Japanese average (*Our World in Data, 2022*). Considering that for many vaccinated individuals the elapsed time since the second vaccination was less than three months at the time of the event (two-dose vaccination coverage on May 29, 2021: 3% based on the Japanese average *Our World in Data, 2022*), the vaccine was assumed to be 80% effective in preventing infection with the Delta variant (*Chemaitelly et al., 2021*). The risk of infection in consideration of vaccination was assessed according to the methodology of a previous study (*Murakami et al., 2022a*).

### Event 1 (additional infection control measure scenario)

With reference to Supersonic (September 18–19, 2021) (*Supersonic, 2021*), an outdoor music festival with thorough infection control measures held in Japan, we evaluated the risk of infection under a hypothetical situation in which Event 1 was held with the addition of further infection control measures:

(a) Antigen testing: By conducting qualitative antigen testing for all participants on the day of the event, we reduced $P_0$ by assuming that asymptomatic infectors who tested positive would be excluded from the event (*Murakami et al., 2022a*).

(b) Distance: The distance from people during the entry, exit, and rest was set to 1.5 m and the distance from people during the attendance of live performances was set to 1 m. The number of people in front of the infector during the attendance of one live performance changed from three to one.

Murakami et al. (2022), *PeerJ*, DOI 10.7717/peerj.13846

**Table 2  Pathways of infection by behavioral pattern.**

| Behavioral pattern | Type of people exposed | Pathway | Note |
|---|---|---|---|
| (A) Attending live performances | People accompanying the infector | Direct droplet spray, direct inhalation of inspirable particles, and inhalation of respirable particles via air | The distance between the infector and the accompanying people or people in front of the infector was as follows: 0.5 m (base scenario), 1 m (distance measure scenario) |
| | | | Frequency of talking of the infector: 0.2 per minute (base scenario), 0.03 per minute (talk measure scenario) |
| | | | The probability that an infector faces each accompanying person and the people in front was 15% and 70%, respectively. |
| | | | The probability that the accompanying person faces the infector was 50%. |
| | People in front of the infector at live performance venues | Direct inhalation of inspirable particles and inhalation of respirable particles via air | |
| | People exposed in restrooms, people exposed at concession stands, and others | Inhalation of respirable particles via air | |
| (B) Entering, exiting, and resting | People accompanying the infector | Direct droplet spray, direct inhalation of inspirable particles, and inhalation of respirable particles via air | The distance between the infector and the accompaniers was as follows: 0.5 m (base scenario), 1.5 m (distance measure scenario) Frequency of talking of the infector: 0.2 per minute |
| | | | The probability that an infector faces each accompanying person was 50%. |
| | | | The probability that the accompanying person faces the infector was 50%. |
| | People in front of the infector at live performance venues, people exposed in restrooms, people exposed at concession stands, and others | Inhalation of respirable particles via air | |
| (C) Using restrooms | People exposed in restrooms | Hand contact | The person touches the contaminated surface two minutes after the virus was deposited on the surface. The exposure from fingers-to-face contact was considered to be 6 h. |
| | | | Frequency of talking of the infector: 0 per minute. |
| | | | Handwashing measures inactivate the virus on fingers. |
| | | | Wearing a mask reduces the frequency of touching the facial mucosal membranes. |

Murakami et al. (2022), *PeerJ*, DOI 10.7717/peerj.13846

*Peer*J

**Table 2** (*continued*)

| Behavioral pattern | Type of people exposed | Pathway | Note |
|---|---|---|---|
| (D) Ordering at concession stands | People exposed at concession stands | Hand contact | The person touches the contaminated surface 1 min after the virus was deposited on the surface. The exposure from fingers-to-face contact was considered to be 6 h. |
| | | | Frequency of talking of the infector: 1 per minute. By considering the talk time to be 10 s, the amount of virus emitted by talking was assumed to be 1/6th of that per minute. |
| | | | Disinfection measures inactivate the virus on surfaces. |
| | | | Wearing a mask reduces the frequency of touching the facial mucosal membranes. |
| (E) Eating | People accompanying the infector | Direct droplet spray, direct inhalation of inspirable particles, and inhalation of respirable particles via air | The distance between the infector and the accompanying people was as follows: 0.5 m (base scenario), 1.5 m (distance measure scenario) |
| | | | Frequency of talking of the infector: 0.2 per minute (base scenario), 0.03 per minute (talk measure scenario) |
| | | | The probability that an infector faces each accompanier was 50%. |
| | | | The probability that the accompanying person faces the infector was 50%. |
| | | | People do not wear masks during meals. |
| | People in front of the infector at live performance venues, people exposed in restrooms, people exposed at concession stands, and others | Inhalation of respirable particles via air | |

**Table 3  Dose by type of person exposed.**

| Types of people exposed | Dose |
|---|---|
| (1) People accompanying the infector | (A) Attending live performances: (direct droplet spray + direct inhalation of inspirable particles + inhalation of respirable particles via air) ×6<br>(B) Entering, exiting, and resting: (direct droplet spray + direct inhalation of inspirable particles + inhalation of respirable particles via air) ×6<br>(E) Eating: (direct droplet spray + direct inhalation of inspirable particles + inhalation of respirable particles via air) ×2 |
| (2) People in front of the infector at live performance venues | (A) Attending live performances: (direct inhalation of inspirable particles) ×1 + (inhalation of respirable particles via air) ×6<br>(B) Entering, exiting, and resting: (inhalation of respirable particles via air) ×6<br>(E) Eating: (inhalation of respirable particles via air) ×2 |
| (3) People exposed in restrooms | (A) Attending live performances: (inhalation of respirable particles via air) ×6<br>(B) Entering, exiting, and resting: (inhalation of respirable particles via air) ×6<br>(C) Using restrooms: (hand contact) ×1<br>(E) Eating: (inhalation of respirable particles via air) ×2 |
| (4) People exposed at concession stands | (A) Attending live performances: (inhalation of respirable particles via air) ×6<br>(B) Entering, exiting, and resting: (inhalation of respirable particles via air) ×6<br>(D) Ordering at concession stands: (hand contact) ×1<br>(E) Eating: (inhalation of respirable particles via air) ×2 |
| (5) Others | (A) Attending live performances: (inhalation of respirable particles via air) ×6<br>(B) Entering, exiting, and resting: (inhalation of respirable particles via air) ×6<br>(E) Eating: (inhalation of respirable particles via air) ×2 |

(c) Mask-wearing: The mask-wearing proportion of the participants was set to 100%.

(d) Restriction of talking during the attendance of live performances and meals: The frequency of talking during the attendance of live performances and meals was set to 0.03 per minute.

(e) Disinfection: Disinfection after ordering at concession stands reduces the viral concentration on surfaces to 1/1,000 (*Murakami et al., 2021*).

(f) Handwashing: Washing hands after using the restroom reduces the viral concentration on fingers to 1/100 (*Murakami et al., 2021*).

(g) Vaccination: The vaccination coverage of the participants was set to 100%. In this case, $P_0$ did not change.

(h) All measures (a–f) are implemented. (i) All measures (a–g) are implemented.

In addition, with measure (h) in place, analyses were conducted under conditions in which the number of participants or $P_0$ was reduced from the base scenario to 75%, 50%, 25%, and 10%.

### Event 2

We performed the risk assessment for Event 2 according to previously reported conditions (*Suñer et al., 2022*). It was reported that appropriate control measures were taken at Event 2; however, the distance measure (b) was not applied (*Suñer et al., 2022*). We therefore considered the above control measures other than distance (b). There were differences from the parameter settings described above with respect to mask-wearing (c) and vaccination (g). Participants were provided with non-woven masks, and 75% of participants reported wearing masks at all or most of the time during the event (*Suñer et al., 2022*). We therefore set the mask-wearing proportion (c) at 75%. Regarding the COVID-19 immunity status, 23% were fully protected (*i.e.*, had received the two-dose vaccination or one-dose vaccine among individuals with a history of COVID-19 infection) and 44% were partially protected (*i.e.*, either one-dose vaccination, two-dose vaccination <14 days before the event, or a history of COVID-19 infection without a vaccine) (*Suñer et al., 2022*). Therefore, as in the previously reported definition (*Suñer et al., 2022*), we assumed that individuals who were immune because of a history of COVID-19 infection are equivalent to vaccinated individuals, and set the vaccination coverage at 67%. The vaccine effectiveness was set at 40% in accordance with the value for the Delta variant among individuals $\geq$ 14 days after the one dose of the vaccination (*Chemaitelly et al., 2021*).

## RESULTS

### Model validation

The total number of already- and newly-infected individuals, who participated in Event 1, was 24.8 (95% uncertainty interval [UI]: 9.2–48.1), 47.0 (95% UI: 12.5–185.5), and 172.7 (95% UI: 25.1–610.0) for those with a 10-, 100-, and 1,000-fold increase in the Delta-variant viral concentrations relative to the wild-type strain, respectively (Fig. 2). These results are in agreement with the reported number of infected cases (45). Under a 100-fold viral concentration and mask-wearing proportion ranging from 0% to 100%, the total number of infected individuals varied from 73.0 (95% UI: 14.7–348.1) to 25.5 (95% UI: 9.6–48.9; Fig. 3).

The infection risk in Event 2 (*i.e.*, the rate of new infections due to the event) was $1.98 \times 10^{-2}$ (95% UI: $0.55 \times 10^{-2}$–$6.39 \times 10^{-2}$; Fig. 4). This was comparable to the estimated value ($1.25 \times 10^{-2}$) in the previous report (*Suñer et al., 2022*).

### Control measure effectiveness

When additional measures were implemented individually at the hypothetical Event 1, the number of newly-infected individuals significantly decreased by vaccination (69%), mask-wearing (57%), and disinfection (54%), and the risk of infection was greatly reduced by implementing all the control measures (all measures except for vaccination: 81%; all measures including vaccination: 94%; Fig. 5). When all measures, except for vaccination,

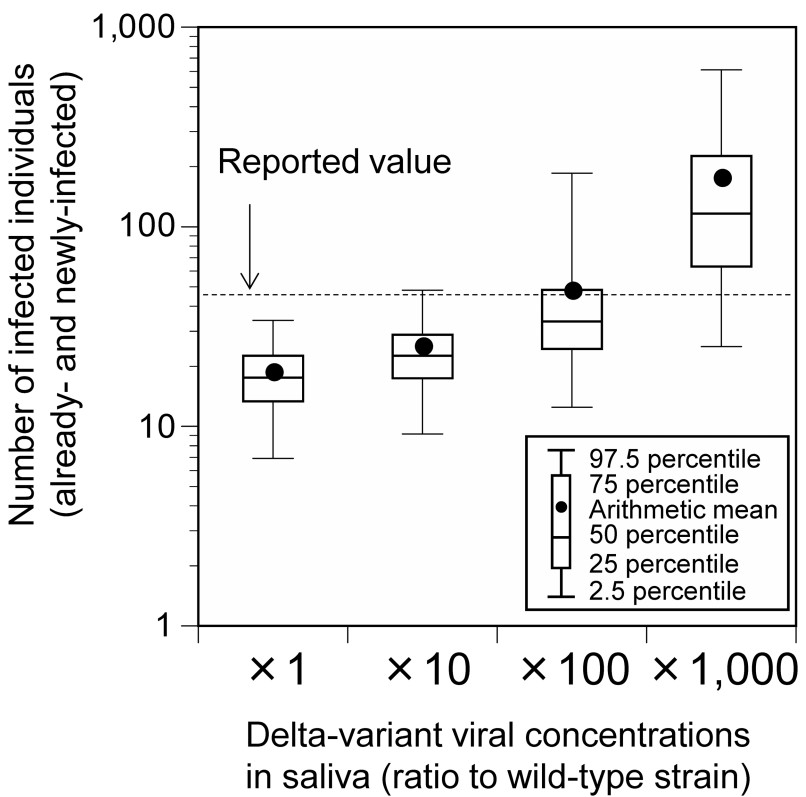

**Figure 2** **Comparison of the estimated and reported numbers of already- and newly-infected individuals (base scenario; Event 1).** Already-infected individuals represent those who were infectors at the time they participated in the event.

were implemented and the number of participants or $P_0$ was reduced, the number of newly-infected individuals was linearly related to the reduction ratio of the number of participants or $P_0$ (Fig. 6). The average number of newly-infected individuals per an infector who attended the event (including those who tested positive) ranged from 0.73 to 0.76, irrespective of the scenarios. If the event organizer considered keeping the number of newly-infected individuals below five as the arithmetic mean and below 10 as the 97.5 percentile, the number of participants or $P_0$ had to be less than or equal to 50% of the base scenario.

## DISCUSSION

### Model validation

In this study, the number of infected individuals or infection risk was estimated using an environmental exposure model for outdoor music festivals, where the number of infected individuals or infection rates has been reported. The reported value at Event 1 in Japan was in the range of 95% UI of the total estimated number of infected individuals at any condition (10-, 100-, and 1,000-fold increase of the Delta-variant concentrations relative to the wild-type strain). It agreed well with the arithmetic mean of the values obtained

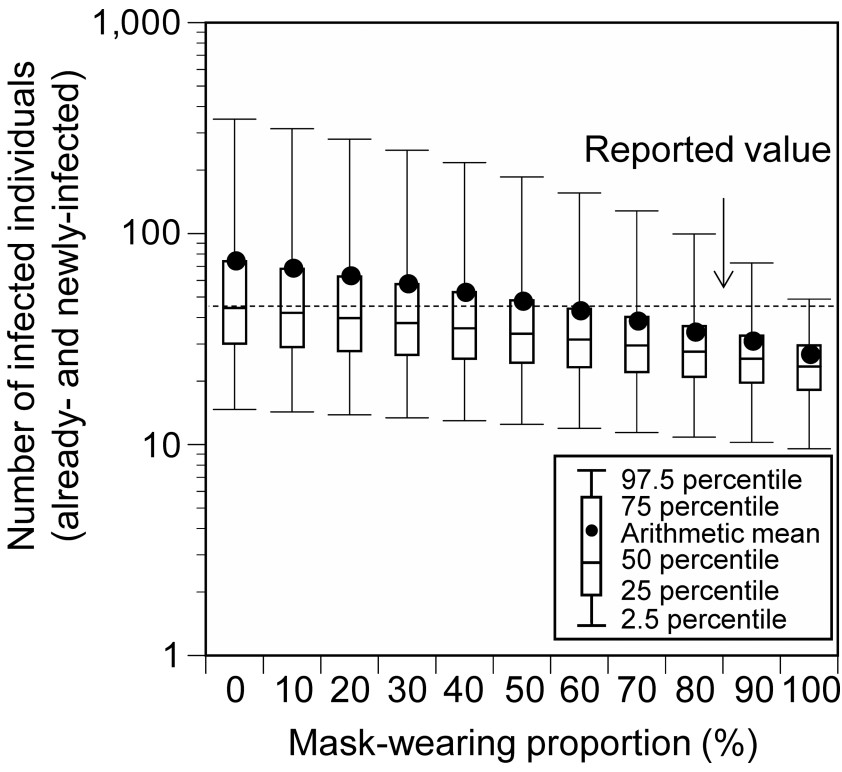

**Figure 3** **Comparison of the estimated and reported numbers of already- and newly-infected individuals under conditions with varying mask-wearing proportions (Event 1).** Viral concentration in the saliva: 100-fold increase relative to the wild-type strain. No additional measures (base scenario).

for the condition with the 100-fold increase in the viral concentration. The results of the sensitivity analysis with varying mask-wearing proportions also showed that the reported value was within the range of the estimates. The reported number of infected individuals might have been underestimated because not all the participants were tested. Based on the information from the free PCR testing that was conducted in the Aichi Prefecture and Nagoya City (eight positive cases among 658 people *Asahi , 2021*), the number of infected individuals was determined to be 90. This value was within the 95% UI of the number of infected individuals under conditions in which the viral concentration was 100 or 1,000 times higher. Considering that the viral loads of the Delta-variant strain are approximately 1,000 times higher than those of the wild-type strain (*Li et al., 2022*), these results support the validity of the infection risk assessment using the environmental exposure model.

Furthermore, regarding Event 2, which took place in Spain, the risk of infection calculated by the model in this study was also similar to the reported value in a previous epidemiological study (*Suñer et al., 2022*). These results highlighted the validity of the model, as the risk assessments performed for the two events in different countries were comparable to the reported values.

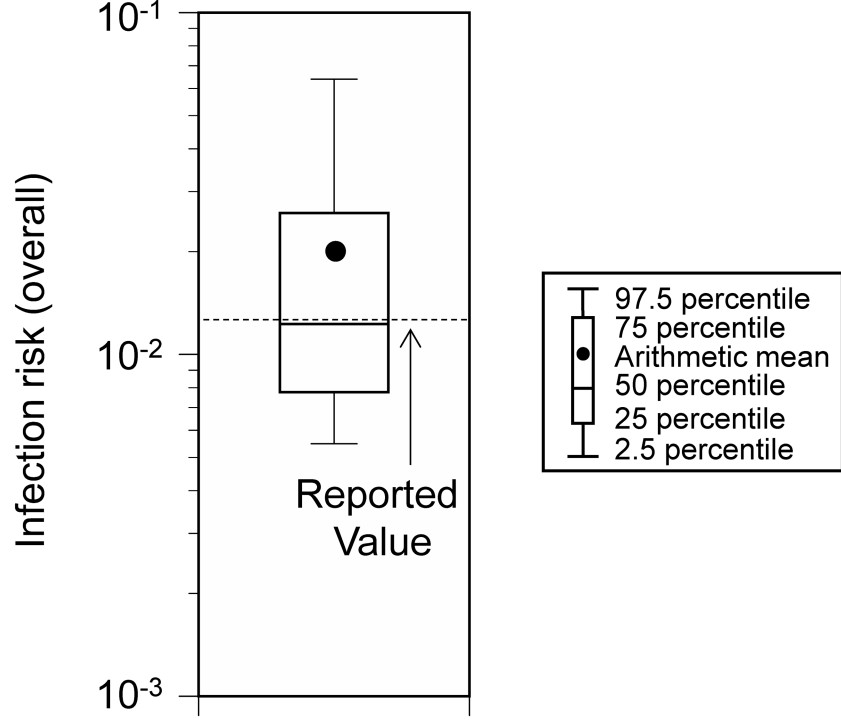

**Figure 4** **Comparison of the estimated and reported infection risk due to the participation in Event 2.** Viral concentration in the saliva: 100-fold increase relative to the wild-type strain.

## Control measure effectiveness and implications

We evaluated the extent to which the risk could be reduced by strengthening the infection control measures at the hypothetical Event 1. Among the additional individual measures, vaccination, mask-wearing, and disinfection of surfaces were effective. Previous epidemiological studies have presented the effectiveness of individual measures and national interventions such as lockdowns in reducing the spread of infection, and have reported that individual measures, especially mask-wearing could reduce the infection risk (*Abaluck et al., 2022*; *Haug et al., 2020*; *Riley et al., 2022*). While it has been suggested that disinfection is not sufficient to reduce the infection risk (*Haug et al., 2020*), *Wang et al. (2020)* reported that disinfection in the households reduced secondary transmission of SARS-Cov-2 within the family by 77%. This study suggested that disinfection could be also effective in reducing the infection risk at mass gathering events, where contact transmission between large numbers of unspecified people occurs. The reduction of the infection risk by mask-wearing and vaccination at mass gathering events has been reported in previous epidemiological studies conducted in the United States (*Sami et al., 2022*) and Spain (*Suñer et al., 2022*). This study consistently demonstrated the large risk reduction effectiveness of these two measures, which are considered to be important for infection risk control at mass gathering events regardless of the country in which it is exercised. As shown in Fig. 1, previous studies included various types of events (*e.g.,* sports, music, meeting, religious events) and study designs (*e.g.* simulations, epidemiological studies) (*Heese et al., 2022*; *Murakami et al.,*

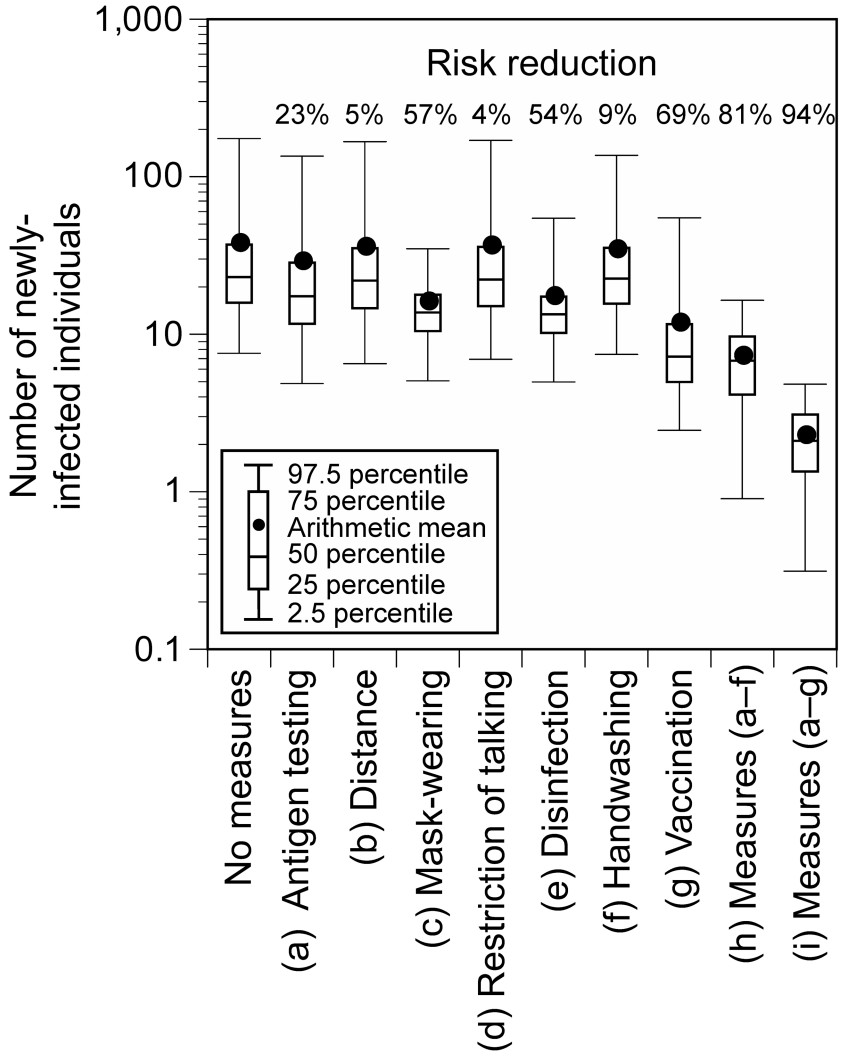

**Figure 5** **Number of newly-infected individuals and risk reduction when additional measures were applied to the base scenario (hypothetical Event 1).** Viral concentration in the saliva: 100-fold increase relative to the wild-type strain.

*2021*; *Revollo et al., 2021*; *Sami et al., 2022*; *Shukla et al., 2021*; *Smith et al., 2022*; *SRLF Trial Group, 2022*; *Suñer et al., 2022*; *Yasutaka et al., 2022*). Previous epidemiological studies could assess the effectiveness in reducing the risk of acquiring infections among exposed individuals but were not able to evaluate the effectiveness in preventing the spread of infection by viruses emitted from already infected individuals (*Murakami, 2022*). This study provided new findings regarding the effectiveness of mask-wearing for both these cases. Furthermore, the combination of all measures resulted in a higher risk reduction (all measures excluding vaccination: 81%; all measures including vaccination: 94%). Thus, the infection risk can be reduced by blocking all pathways of virus transmission including direct exposure, direct inhalation, contact, and air inhalation.

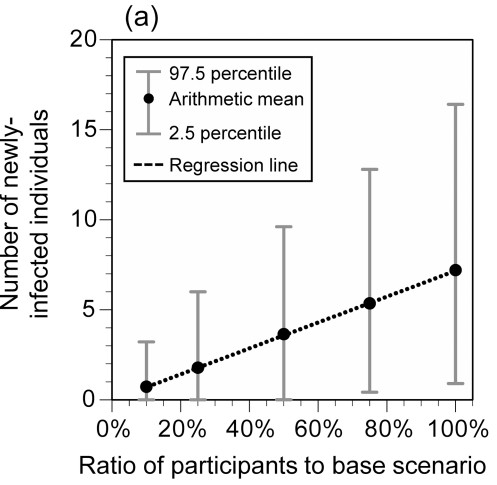
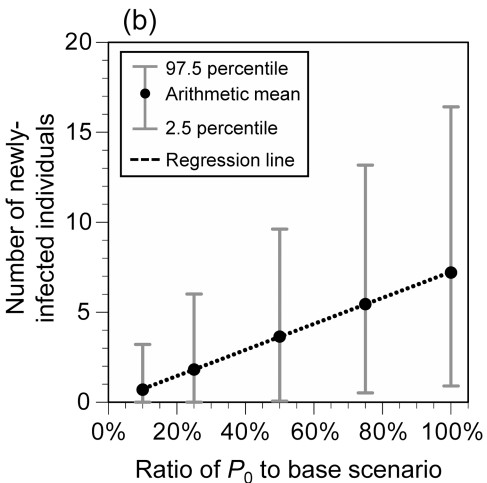

**Figure 6 Number of newly-infected individuals for varying ratios of the number of participants (A) and $P_0$ (B) to the base scenario (hypothetical Event 1).** $P_0$ crude probability of a participant being an infector. Viral concentration in the saliva: 100-fold increase relative to the wild-type strain. Additional measures (A–F) were implemented. When the number of participants was 10% (739), the sum of infectors, people accompanying the infector, people in front of the infector at live performance venues, people exposed in restrooms, and people exposed at concession stands exceeded the number of participants in seven of 10,000 simulations. The number of newly-infected individuals in these runs was calculated by summing the number of newly-infected individuals calculated for each group and dividing it by the total number of participants (739).

During mass gathering events, the extent to which any measures are implemented depends on the organizers' decisions or society's consensus on how many newly-infected individuals are acceptable. For example, in this study, the number of newly-infected individuals at the hypothetical Event 1 was estimated to be 7.2 (95% UI: 0.9–16.4) even if all measures, except for vaccination, were implemented. If the benchmark of acceptable newly-infected individuals was set to less than five and 10 as the arithmetic mean and 97.5 percentile, respectively, additional measures would be necessary such as allowing only vaccinated people to participate or limiting the number of participants to less than or equal to 50%. In addition, although the infection status fluctuates from time to time, there is a linear relationship between $P_0$ and the number of newly-infected individuals, which makes it possible to determine whether additional measures are necessary for holding mass gathering events or whether to refrain from holding such events.

## Uncertainty and limitations

This study has some sources of uncertainty. First, $P_0$ was set from estimates based on reported values for infection rates in the host location. This number may be underestimated because several asymptomatic infectors may have not been identified. In this regard, however, $P_0$ at Event 2 used in this study was similar to the percentage of persons presumed to have already been infected at the event in the previous report (*Suñer et al., 2022*). Second, the risk reduction due to 100% vaccination measures (Fig. 5) may be underestimated, because vaccinated individuals are considered to have a lower probability

of being infected than unvaccinated individuals and thus possibly yield a lower $P_0$. Third, consistent with previous other studies (*Jones, 2020*; *Murakami et al., 2022a*; *Murakami et al., 2021*; *Yasutaka et al., 2022*; *Zhang et al., 2022*), we used the dose–response equation based on SARS-CoV in mice. This parameter was similar to that for SARS-CoV-2 obtained from ferrets and the estimated human exposure (*Zhang & Wang, 2021*). The estimated infection risk was slightly lower than the infection risk observed in the SARS-CoV-2 human challenge (*Killingley et al., 2022*); the risk of infection at 55 focus forming unit was 53% in the human challenge, whereas it was 25% (95% UI: 15–48%) in this study. Fourth, while information on the proportion of adherence to mask-wearing control measures was available for Event 2, similar details for Event 1 were not available. Therefore, we conducted a sensitivity analysis using 50% as the base scenario and varying mask-wearing proportions. Fifth, for Event 1, we assumed a 45% vaccination coverage and 80% vaccine effectiveness based on the two-dose vaccination status. The infection risk might be slightly overestimated, because 5% of the individuals received one dose of the vaccination $\geq 14$ days before the event (*Our World in Data, 2022*). Similarly, in Event 2, the vaccination coverage was set at 67% (sum of 44% partially protected and 23% fully protected) based on COVID-19 immunity status, and the vaccine effectiveness was 40% based on the value for those who were partially protected. The risk of infection at Event 2 might also be overestimated, as the vaccine effectiveness among fully protected individuals may be higher than 40% (*Chemaitelly et al., 2021*).

This study has several limitations. First, the risk of infection outside the event was not assessed in this study; however, confirmed infected individuals may have been infected during activities outside the event. In particular, those who accompany infectors might also act together, even outside the event. Second, we assessed the risk of infection with the Delta variant but did not consider the Omicron variant or any new variants that might arise thereafter. Updated changes in viral concentrations (*Salvagno et al., 2022*) and vaccine effectiveness (*Andrews et al., 2022*), as we have done in this study, are promising with regard to accommodating risk assessment for new variants. Further findings on the parameters regarding these variants are needed to address them. Third, we validated the model based on the total number of infected individuals or the infection rate but did not validate the detailed calculations within the model such as the exposure rates related to each infection pathway and the risk of infection for each type of exposed person. Case-control studies with behavioral records of event participants and environmental measurements of viral concentrations in the air and surface would fill these knowledge gaps.

Despite these limitations, a model for outdoor music festivals was successfully developed in this study and its validity was evaluated. The results of this study guide decision-making related to event organization such as the need to implement additional measures.

## NOTES

This article has already been registered for Preprints on medRxiv (*Murakami et al., 2022b*). DOI is as follows: https://doi.org/10.1101/2022.02.28.22271676.

## ACKNOWLEDGEMENTS

We would like to thank Editage for English language editing and Dr. Kotoe Katayama and Dr. Masaaki Kitajima for their advice.

### Funding

The authors received no funding for this work.

### Competing Interests

Tetsuo Yasutaka reports a relationship with Kao Corporation, Nippon Professional Baseball Organization, Yomiuri Giants, Tokyo Yakult Swallows, the Japan Professional Football League, and the Japan Professional Basketball League: funding grants.

There are no patents to disclose.

This study was conducted as part of a comprehensive research project, comprising members from two private companies, Kao Corporation and NVIDIA Corporation, Japan. No authors in this study belong to these companies. Michio Murakami, Seiya Imoto, and Tetsuo Yasutaka have attended the new coronavirus countermeasures liaison council jointly established by the Nippon Professional Baseball Organization and Japan Professional Football League as experts without any rewards. Tetsuo Yasutaka is an advisor to the Japan National Stadium and Japan Professional Football League. Other authors declare no competing interests.

### Author Contributions

- Michio Murakami conceived and designed the experiments, performed the experiments, prepared figures and/or tables, authored or reviewed drafts of the article, and approved the final draft.
- Tsukasa Fujita performed the experiments, analyzed the data, prepared figures and/or tables, authored or reviewed drafts of the article, and approved the final draft.
- Pinqi Li performed the experiments, analyzed the data, authored or reviewed drafts of the article, and approved the final draft.
- Seiya Imoto conceived and designed the experiments, authored or reviewed drafts of the article, and approved the final draft.
- Tetsuo Yasutaka conceived and designed the experiments, performed the experiments, prepared figures and/or tables, authored or reviewed drafts of the article, and approved the final draft.

### Data Availability

The data are available as a Supplementary File.

### Supplemental Information

Supplemental information for this article can be found online at http://dx.doi.org/10.7717/peerj.13846#supplemental-information.

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
