# Peer review of "Development of a COVID-19 risk assessment model for participants at outdoor music festivals: evaluation of the validity and control measure effectiveness based on two actual events in Japan and Spain"

_PeerJ, doi:10.7717/peerj.13846_

## Round 0.1 · original submission · Major Revisions

Kindly reply to reviewers on point by point basis.

Reviewer 1 ·

Basic reporting

This work pertains to a small-scale covid risk assessment study of a group of people. Overall the work is clear and unambiguous. A group of people has been for COVID infection when without precautionary measures. The outbreak has been modeled cluster outbreak model. The study claims to have achieved a 94% reduction in the cases if proper measures are put into place.
1. The introduction is weak.
2. How, this study can be helpful to other areas with similar settings is also not properly mentioned.
3. This is one such study, how it can be globally applied has not been discussed in detail.
4. Modeling approach is not clear. The methodology behind it is not clear.
5. This work is just reporting its results. The discussion needs to be discussed in reference to relevant published work. It must not only stick its results to delta strain. What about other strains, e.g., omicron
6. As a global pandemic, the results of this study must be applicable to the majority of regions where COVID is prevalent.
7. Uncertianities of this work, should be placed as a separate section.

Experimental design

The experimental design is very restricted. It is still not clear how a separate secluded event can be used for the global management of this pandemic. Either, the authors need to substantiate their claims, or there must be others ways to validate this work.
Some more Statistical techniques need to be used. Why this model has been used may kindly be discussed.
In the model development section, a reader is confused, about what actually has been used from the statistical techniques. It is requested to simplify this section.
Overall methods sections needs to be elaborated.

Validity of the findings

The results need to be validated with some separate random events. It is very important. It is requested to keep a global perspective into consideration. Studies from Asia, South Asia, and the Asia Pacific as well as from Europe and the Americas can be used to discuss. As said, methods sections need to be elaborately discussed.
Further, the discussion section has to include global studies and the results must be validated/ discussed with those.

Additional comments

Overall, the work requires substantial restructuring and additions, to stand as a global level COVID management model study

·

Basic reporting

The reporting of the study is good. Overall the work is a case study, wherein participants have been in a festival and with and without measures, their vulnerability to covid infection has been determined. The conduct of the study of the fine. However, I have reservations about the novelty of the work.

Experimental design

The design of the study has some issues. Can a single event consisting of over 7000 people be used to prepare a risk model for a pandemic such as COVID needs to be substantiated with similar works from other parts of the world. As a single event the design of the experiment is clear and understandable.

Validity of the findings

Again, as far as the single event findings are concerned can be used to model the COVID-19 pandemic is questionable. COVID -19 depends has been observed to depend on various parameters about what has been used to evaluate the risk in this study. Therefore the veracity of the usefulness of this study needs to be substantiated by other global studies similarly carried out in other parts of the World. The discussion needs to be very strong substantiated by the facts and figures from other published work.

Additional comments

The title is very simple, "Development of a COVID-19 risk-assessment model for participants at an outdoor music festival: Evaluation of the validity and control measure effectiveness".

You need to revise it so that it reflects that this is a study that has global implications. Overall huge emphasis needs to be drawn on the usefulness of this study for other regions of the world, taking into consideration all the factors that govern the risk of a population to covid-19

---

## Round 0.2 · Minor Revisions

Dear Authors,

Thank for taking care of the suggestions provided by the reviewers. I appreciate your replies. However, I still feel that, this work can be significantly improved in quality and content, if the geospatial aspect of the problem is added to this manuscript.

For this I would suggest, to kindly add two maps:

1. A map showing the location of the Japan and Spain (with locations of the reported events shown using points) vis-a-vis their relation with the World.

2. If possible, if there are any similar studies been carried out in other parts of the world, a map depicting their location and key results, would be very meaningful and significant addition to this work. You can discuss such works in your discussion section as well.

Some of my recent works can be used for understanding the map visualization, I intend from your work. My works need not to be cited, I am referring to them for your reference purposes only.

I am looking forward for the revised version of your manuscript.

Best,
Gowhar Meraj

---

## Round 0.3 · accepted · Accept

Thank you for incorporating all the suggested changes. Congratulations once again for this wonderful work

Reviewer 1 ·

Basic reporting

The manuscript has been improved a lot. The analysis of two case studies, Japan and Spain has added huge value to its content

Experimental design

The design of the experiment is acceptable

Validity of the findings

The results are well validated.

Additional comments

I recommend its acceptance in current form

·

Basic reporting

The authors have considered all the comments, I had previously asked them to incorporate.

Experimental design

Using two case studies, the manuscript has significantly improved in experimental design

Validity of the findings

The results can be now applied to other areas of the Globe for pandemic/ epidemic control measure.

Additional comments

I thank and congratulate the authors for working out on their manuscript and submitting the revised version.